# Comprehensive Analysis of Oligo/Polysialylglycoconjugates in Cancer Cell Lines

**DOI:** 10.3390/ijms23105569

**Published:** 2022-05-16

**Authors:** Masaya Hane, Ken Kitajima, Chihiro Sato

**Affiliations:** 1Bioscience and Biotechnology Center, Nagoya University, Nagoya 464-8601, Japan; mhane@agr.nagoya-u.ac.jp (M.H.); kitajima@agr.nagoya-u.ac.jp (K.K.); 2Graduate School of Bioagricultural Sciences, Nagoya University, Nagoya 464-8601, Japan; 3Integrated Glyco-Biomedical Research Center (iGMED), Institute for Glyco-Core Research (iGCORE), Nagoya University, Nagoya 464-8601, Japan

**Keywords:** oligosialic acid, polysialic acid, non-human sialic acid, ganglioside, cancer

## Abstract

In cancer cells, cell-surface sialylation is altered, including a change in oligo/polysialic acid (oligo/polySia) structures. Since they are unique and rarely expressed in normal cells, oligo/polySia structures may serve as promising novel biomarkers and targets for therapies. For the diagnosis and treatment of the disease, a precise understanding of the oligo/polySia structures in cancer cells is necessary. In this study, flow cytometric analysis and gene expression datasets were obtained from sixteen different cancer cell lines. These datasets demonstrated the ability to predict glycan structures and their sialylation status. Our results also revealed that sialylation patterns are unique to each cancer cell line. Thus, we can suggest promising combinations of antibody and cancer cell for glycan prediction. However, the precise prediction of minor glycans need to be further explored.

## 1. Introduction

Every cell surface is covered by “glycocalyx”. The glycocalyx consists of glycoproteins, proteoglycans, and glycolipids, and it regulates cellular events, such as cell–cell interactions and viral and bacterial infections. Alterations in cell-surface glycosylation in several diseases are well known and have been previously reported [1]. Glycosylation is associated with the malignant state of tumors. In addition, several glyco-epitopes, especially sialyl glyco-epitopes such as sialyl Lewis A (sLe^a^, CA19-9) and sialyl Lewis X (sLe^X^, SLX), are generally used as biomarkers. Almost all deuterostomes, including vertebrates, contain sialic acid (Sia) at the non-reducing terminus of their glycans. Therefore, sialoglycoconjugates exist in the outermost part of cells and play important roles in cellular functions. Sia structures are categorized into three major groups based on the modification of the C-5 position: *N*-acetylneuraminic acid (Neu5Ac), *N*-glycolylneuraminic acid (Neu5Gc), and deaminoneuraminic acid (Kdn). In vertebrates, two Sia species, Neu5Ac and Neu5Gc, are mainly expressed. However, human cells, which lack functional CMP-Neu5Ac hydroxylase (CMAH), only synthesize and express Neu5Ac. Nevertheless, the occurrence of Neu5Gc and KDN-containing glycoconjugates in human cancer cells has been reported [2,3,4]. In humans, Neu5Gc and Kdn are considered to be derived from food and used for glycosylation in the salvage pathway. In addition, a relationship between hypoxia and de novo synthesis of Kdn and an increase in Neu5Gc incorporation into glycoconjugates has been reported [5,6]. Moreover, on rare occasions, Sia residues are linked to each other and form oligomerized (2 to 7 Sia residues)/polymerized (≥8 Sia residues) forms, oligosialic acid (oligoSia/OSA), and polysialic acid (polySia/PSA) on glycoproteins and glycolipids (gangliosides). Oligo/polySia structures are biosynthesized by ST8 α-*N*-acetyl-neuraminide α2,8-sialyltransferase (ST8SIA). Humans have six ST8SIA enzymes (ST8SIA1, 2, 3, 4, 5, and 6), and evidence indicates that these enzymes and oligo/polySia structures that are produced are related to tumor malignancy [7,8,9,10]. For example, ST8SIA1 and diSia ganglioside GD3 have been well studied in cancer [11,12]. In addition, the diversity of Sia modifications, including sulfation, acetylation, and lactylation, is well known [7]. Evidence of the relationship between Sia modification and cancer has accumulated, but its biosynthetic pathways and functions remain largely unknown [13]. Sialoglycoconjugates are closely related to cancer; therefore, a comprehensive analysis of cancer cell-surface sialoglycoconjugates is needed to explore new biomarkers and therapeutic targets.

In this study, we first predicted the cell surface glycans of human cancer cell lines, particularly oligo/polySia structures, using GlycoMaple [14,15] and depmap RNAseq datasets. Second, we analyzed cell surface sialoglycoconjugates with anti-sialoglycoconjugate antibodies [7,16] using flow cytometry. Next, we verified the consistency between observed and predicted sialoglycoconjugates (Figure 1).

## 2. Results

### 2.1. Bioinformatic Analysis

To predict the cell surface oligo/polysialoglycoconjugate structure, we obtained glycan-related gene expression datasets from the depmap portal (Table 1). We visualized and mapped the glycan structures using GlycoMaple software (Appendix A). The predicted glycan structures include complex capping *N*-glycans/*O*-glycans/GSLs, mucin-type *O*-glycans, and ganglioside biosynthesis. Oligo/polySia structures (12E3/A2B5 related epitope) were predicted in cancer cell lines; HEL, IMR-32, SK-N-SH, U-251MG, A549, and MCF-7 cells; and MOLT-3, Caki-1, SK-MEL-2, PA-1, and PC-3. DiSia2,3-Gal structures (S2-566/FS1/FS3 related epitope) were estimated in A549, IMR-32, MCF-7, and SK-MEL-2; strongly in U-251MG, HEL, SK-N-SH, SW-13, Caki-1, MIA PaCa-2, and HeLa; and weakly in PA-1 (Figure 4A). We predicted oligo/polySia structures using ST8SIAs and their known substrate expression data (Figure 4B,C). PolySia structures were predicted only in IMR-32 and SK-N-SH cells (Figure 4C).

### 2.2. Flow Cytometry and Western Blotting

We analyzed cell surface sialoglycoconjugates by flow cytometry using eight defined anti-sialoglycoconjugate antibodies (Table 2 [17,18,19,20,21,22,23,24,25,26], Figure 2). Oligo/polySia (Neu5Ac) structures are recognized by 12E3 [17]. MOLT-4, IMR-32, SK-N-SH, U251-MG, HepG2, A549, SW-13, LS174T, and SK-MEL-2, and HeLa cells were 12E3 positive. Two neuroblastoma IMR-32 and SK-N-SH cells were strongly positive and expressed a polysialylated neural cell adhesion molecule (polySia-NCAM); however, the other cells did not express detectable polySia-NCAM by Western blotting (Figure 2Q). A2B5 recognizes triSia (Neu5Ac), GT3, and OAcGT3 [18,19,20]. All 16 cells expressed the epitope of A2B5 to a certain degree. S2-566 recognizes diSia-Gal, GD3, and GT1b [21]. MOLT-4, HEL, LS174T, and SK-MEL-2 cells were S2-566 positive. FS1 recognizes GD3 and GQ1b [22,23]. MOLT-4, HEL, A549, SW-13, Caki-1, HeLa, and PA-1 cells were FS1 positive. FS3 recognizes GQ1b [22,23]. The HEL, Caki-1, MIA PaCa-2, LS174T, SK-MEL-2, and HeLa cells were FS3 positive. 2-4B recognizes oligoSia (Neu5Gc) [24]. A549, SW-13, Caki-1, LS174T, SK-MEL-2, HeLa, and PA-1 cells tested positive. The addition of excess Neu5Gc to the culture medium enhanced the 2-4B reactivity of HepG2, A549, LS174T, and PA-1 cells. PC-3 cells became 2-4B positive upon the addition of Neu5Gc (Figure 3A). Kdn8kdn recognizes oligoSias (Kdn) [17,25]. HepG2, A549, and PA-1 cells were kdn8kdn positive, and the reactivity of kdn8kdn was enhanced by the addition of excess Kdn to the culture medium. LS174T and SW-13 cells were kdn8kdn-positive upon the addition of excess Kdn (Figure 3B). Sulfated Sia (H_3_OS-8O-Neu5Ac) is recognized by 3G9 [26]. A549, SW-13, Caki-1, LS174T, SK-MEL-2, and MCF-7 cells were 3G9 positive (Figure 2).

## 3. Discussion

Cancer cell glycocalyx is poorly understood even though it may be essential for the development of therapeutic targets. In this study, we analyzed cell-surface sialoglycoconjugates using flow cytometry and predicted the glycan structure of oligo/polySia using glycan gene expression datasets. Our study suggests that sialoglycoconjugates vary in cancer cell lines, and several patterns were observed. Using Western blotting, polySia-NCAM expression was detected in two neuroblastoma cell lines, IMR-32 and SK-N-SH (Figure 2Q). As polySia-NCAM expression is strictly limited to the neural cells in distinct regions of the brain, its expression in the neuroblastoma cell lines is reasonable. Moreover, both cell lines exhibited poor reactivity with anti-ganglioside antibodies despite the high expression levels of genes, such as *ST8SIA1* in IMR-32 (Appendix A). Therefore, polySia-NCAM may inhibit the binding of the ganglioside antibodies or the presence of competition with the biosynthesis of ganglioside. As the amount of endogenous CMP-Sia is restricted, competitive CMP-Sia usage for polySia and gangliosides may occur, leading to its dominant expression of polySia or gangliosides.

The expression of polySia-NCAM in MOLT-4, A549, HeLa, and LS174T cells was not observed by Western blotting (Figure 2Q), although 12E3 staining was observed by flow cytometry. Therefore, these cells may have oligoSia on their cell surfaces, and gangliosides may be recognized by 12E3. As the epitope of 12E3 was shown to be (Neu5Ac) *n* ≥ 5 and the epitope towards the antibody is usually 2–3-carbohydrate in size, the extended di- or oligoSia-containing ganglioside might be a candidate. The precise epitope is still not fully clear because of the discrepancy in staining using 12E3; however, based on cancer-cell staining, the 12E3 antibody is promising for diagnosis by flow cytometry and for the therapy of several cancers. Almost all cancer cells were stained by A2B5 staining, suggesting that this antibody may be promising for diagnosis.

Based on these results, anti-ganglioside antibodies may be useful for treating melanoma and leukemia. Non-human Sias (Neu5Gc and Kdn) have been detected in several cell lines. Neu5Gc is a major component of bovine serum [27]. In this study, the amount of Neu5Gc incorporated into the cell-surface glycoconjugates and their capacities varied in different cell lines. In particular, A549 and LS174T cells showed high capacities for both Neu5Gc and Kdn. Only PC-3 cells were cultured in the 7% Fetal bovine serum (FBS) medium, and the cells showed 2-4B reactivity after the addition of Neu5Gc. Therefore, Neu5Gc in 7% FBS was not sufficient for 2-4B recognition, but PC-3 may have a large capacity for Neu5Gc. The expression of these non-human Sias in cancer cells has been reported to participate in hypoxia and the de novo synthesis pathway [5,6]. Hypoxia is a hallmark of cancer [28], and a relationship between chronic inflammation and cancer pathogenesis has been reported [6]. Moreover, the high capacity of non-human Sias is an important feature of therapeutic methods. Recently, click chemistry has been developed for cell labeling and therapeutic purposes. As only some cells showed a high capacity for non-human Sias, they can be engineered by click chemistry and tagged as therapeutic-target cells [29,30]. Additionally, the diversity of Sia modifications such as sulfation, acetylation, and lactylation is well known [7]. In this study, we examined the anti-8SNeu5Ac antibody 3G9 and found that its reactivity was similar to that of anti-non-human Sias antibodies. The function of 8-*O*-sulfation of Neu5Ac remains unclear, but it has been reported that 8-*O*-sulfation of Neu5Ac confers resistance to sialidase [31]. Notably, non-human Sias have the same features [27].

GlycoMaple is a powerful tool for predicting entire glycan structures. However, the predicted and experimental oligo/polySia structures were different in this study (Figure 4B, prediction (1) and Figure 4C). When we introduced a list of several genes that are involved in oligo/polySia synthesis into the prediction criteria (Appendix A), the accuracy of the prediction improved (Figure 4B, prediction (2)). In addition, because polySia structures only exist on particular substrates, the co-expression of polysialyltransferases and cognate substrates NCAM and NPR2 should be considered [32]. In this regard, we also considered the expression of substrates shown in Appendix A for the prediction criteria. We predicted that the two neuroblastoma cell lines IMR-32 and SK-N-SH express polySia-NCAM (Figure 4B, prediction (2)). Using flow cytometry, several cell lines were found to be 12E3 positive (Figure 4C); this is likely due to the presence of substrates other than NCAM or the unknown oligosialoglycoconjugates described above. By conducting Western blotting, only IMR-32 and SK-N-SH cells showed positive staining for polySia-NCAM (Figure 4C). Taking these results together, for polySia-NCAM prediction, a combination of ST8SIA2 and NCAM expression is not sufficient; however, the presence of ST8SIA3 is required (Figure 4B, Prediction (2), Appendix A), which is a notable feature. ST8SIA2 and ST8SIA4 are well-known polysialyltransferase; however, the expression of either or both genes did not always lead to that of polySia-NCAM, although we believe that the expression of these two genes is required for that of polySia-NCAM. NCAM expression is an important factor. If there is a low level of NCAM expression, polySia staining cannot be observed, which indicates that polySia is highly restricted to NCAM. However, in this study, we found that the expression of ST8SIA3 may be another important factor in predicting the presence of polySia-NCAM. Therefore, further biochemical analysis is required.

ST8SIA2 and ST8SIA4 can be used to synthesize polySia on substrates other than NCAM. ST8SIA2 and ST8SIA4 synthesized polySia on CADM1 [32] and NRP2 [33], respectively. However, regardless of the presence of CADM1 and ST8SIA2 in PA1, polySia staining was not observed (Figure 2N,Q), which is inconsistent with a previous report that stated that ST8SIA2 synthesizes polySia on CADM1 in vitro [34]. Prediction (1) could not predict the results for PA1 (Figure 4, PA1). In contrast, prediction (2) could predict them successfully. These results indicate that the polysialylation of PA1 is unrelated to the presence of CADM1 in the cell. U-251-MG is an interesting cell line because we could not observe polySia staining, even though it expresses ST8SIA2, ST8SIA4, NCAM, CADM1, and NRP2. Neither prediction (1) nor (2) could predict the absence of polySia (Figure 4B,C; U251-MG). Thus, these cells may have an unidentified biosynthetic mechanism.

With regards to A2B5 epitope prediction, we considered the presence of GT3 and oligoSia on mucin-type glycans because A2B5 was demonstrated to detect triSia-BSA by Western blotting. Therefore, prediction (2) for the A2B5 epitope was substantially improved compared with prediction (1) (Figure 4B). GD3 is recognized by both S2-566 and FS1. However, the reactivities of the two antibodies toward GD3 were different. There may be unknown factors behind this difference. Finally, several cells showed FS3-positive results; however, the prediction of GQ1b was not successful. DiSia and triSia structures are present not only on glycolipids (including b- and c-series gangliosides, respectively), but also on glycoproteins. The recognition mechanism of anti-ganglioside antibodies (S2-566, FS1, FS3, and A2B5) toward the common epitopes diSia and triSia and between proteins and lipids is interesting because the staining patterns between cancer cells with these antibodies are markedly different. These findings may be useful for the diagnosis of cancer.

In conclusion, anti-sialoglycoconjugate antibodies are promising for the diagnosis and treatment of cancer. The proposed prediction strategy, prediction (2), may be useful for the prediction of the oligo/polySia epitope; however, further studies on the epitopes and minor modifications of Sias are necessary to understand the precise nature of cancers.

## 4. Materials and Methods

### 4.1. Materials

Human cancer cell lines (Table 1) including adeno carcinoma cell A549, kidney carcinoma clear cell Caki-1, erythroblast leukemia cell HEL, pancreatic carcinoma cell MIA PaCa-2, T-leukemia cell MOLT-4, neuroblastoma cell IMR-32, prostatic cancer cell PC-3, melanoma cell SK-MEL-2, small-cell carcinoma of the adrenal cortex SW-13, and astrocytoma cell U-251 MG were purchased from the Japanese Collection of Research Bioresources (JCRB) cell bank (Kobe, Japan). The cervical cancer cell line HeLa, hepatoma cell HepG2, breast cancer cell MCF-7, ovarian teratocarcinoma cell PA-1, colon adenocarcinoma cell line LS174T, and neuroblastoma cell SK-N-SH were provided by the RIKEN BioResource Research Center through the National Bio-Resource Project of the MEXT/AMED (Tsukuba, Japan). Eagle’s minimum essential medium (EMEM), RPMI1640, Kaighn’s modification of Ham’s F12 medium (F12K), Leibovitz’s L-15 medium (L-15), and non-essential amino acids (NEAA) were purchased from Wako (Osaka, Japan). Fetal bovine serum (FBS), Alexa647-conjugated goat anti-mouse IgG, and Alexa647-conjugated goat anti-mouse IgM were purchased from Thermo Fisher Scientific (Waltham, MA, USA). Streptomycin sulfate and penicillin G were purchased from Meiji Seika Pharma (Tokyo, Japan). All anti-sialoglycoconjugate antibodies (Table 2) were purified before use. Pre-stained XL-Ladder was purchased from Integral (Tokushima, Japan).

### 4.2. Cell Culture

IMR-32, A549, MIA PaCa-2, HeLa, PA-1, and MCF-7 cells were cultured in EMEM supplemented with 1% NEAA. SK-N-SH, U-251 MG, HepG2, Caki-1, LS174T, and SK-MEL-2 cells were cultured in EMEM. MOLT-4 and HEL cells were cultured in RPMI1640. PC-3 cells were cultured in F12K medium. SW-13 cells were cultured in L-15 medium. All mediums contained 0.5 mg/mL of streptomycin sulfate and 100 units/mL of penicillin G. With the exception of PC-3, all media contained 10% FBS. For PC-3 cells, the medium contained 7% FBS. Except for SW-13 cells, all cells were cultured in a 5% CO_2_ and 95% air humidified atmosphere at 37 °C. SW-13 cells were cultured in a 100% air humidified atmosphere at 37 °C. Neu5Gc or Kdn (0.1 mM) was added to enhance the incorporation of Neu5Gc and Kdn into glycoconjugates.

### 4.3. Flow Cytometric Analysis

For flow cytometry analysis, all cells were cultured as described above and washed with PBS. After washing, all cells were harvested using a scraper and an FCM buffer (0.5% BSA and 5 mM ethylenediaminetetraacetic acid in PBS). The cells were incubated with antibodies on ice for 30 min. After washing, the cells were incubated with Alexa647-conjugated goat anti-mouse IgG or Alexa647-conjugated goat anti-mouse IgM (4 μg/mL) in the FCM buffer. A Gallios flow cytometer (Beckman Coulter, Brea, CA, USA) was used for data collection, and Kaluza software (Beckman Coulter) was used for data analysis.

### 4.4. Western Blotting

Western blotting was performed as previously described [35]. Briefly, 10 μg of protein from each cell lysate was separated by SDS–PAGE (7% acrylamide gel), and proteins were blotted onto a PVDF membrane. The membrane was then blocked with 1% BSA in PBS containing 0.05% Tween 20 (PBST) at 25 °C for 1 h. The membrane was then incubated with the 12E3 antibody. After washing with PBST, the membrane was incubated with peroxidase-conjugated anti-mouse IgG + M secondary antibody (1/5000 dilution) at 37 °C for 1 h.

### 4.5. Bioinformatic Analysis

Gene expression data (Table 1) were obtained from the depmap portal (https://depmap.org/portal/, accessed on 12 May 2022) [36], and glycan expression pathways were predicted using GlycoMaple (https://beta.glycosmos.org/glycomaple/index, accessed on 12 May 2022) [37]. We analyzed the complex capping of *N*-glycans/*O*-glycans/GSLs, mucin-type O-glycans, and biosynthesis of gangliosides using a gene set for each (Appendix A). We analyzed the oligo/polySia structures using a gene set for each (Appendix A).

## Figures and Tables

**Figure 1 ijms-23-05569-f001:**
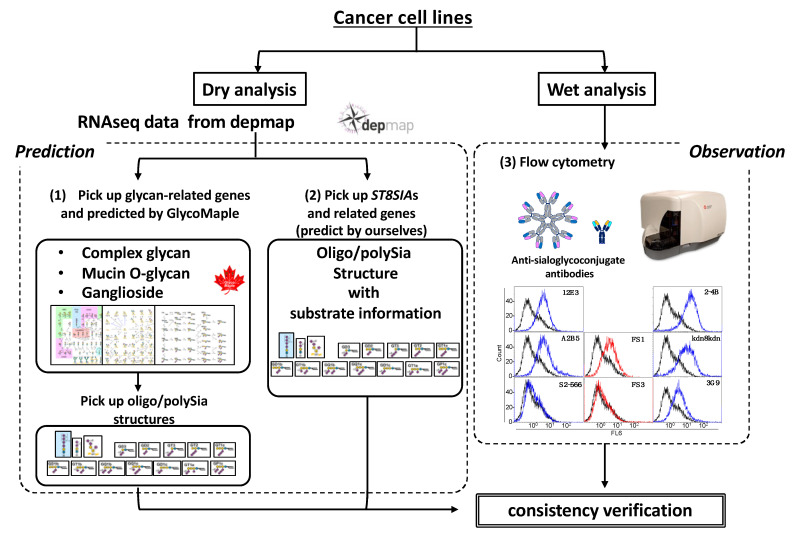
The strategy scheme of this study. For dry analysis, RNAseq data were used to predict oligo/polySia structures by picking up (**1**) glycan-related genes followed by the GlycoMaple and (**2**) ST8Sias and the related genes followed by our own method. For wet analysis, (**3**) flow cytometric analysis of the same cell lines used in dry analysis was performed. Based on the results, the consistencies of the observed and predicted sialoglycoconjugates were verified.

**Figure 2 ijms-23-05569-f002:**
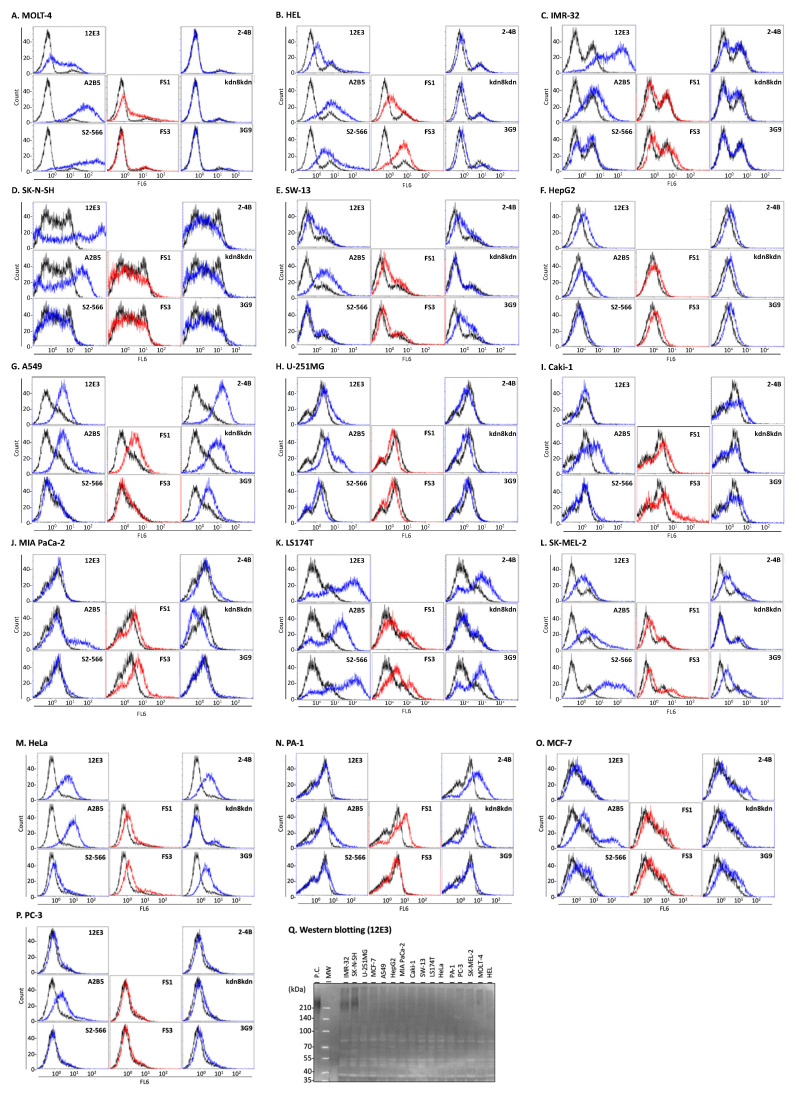
Flow cytometric analysis of cancer cell lines. Human cancer cell lines were analyzed by flow cytometry using eight different anti-sialoglycoconjugate antibodies. Cell surface expression profiles of sialoglycoconjugates are shown in (**A**) MOLT-4, (**B**) HEL, (**C**) IMR-32, (**D**) SK-N-SH, (**E**) U-251MG, (**F**) HepG2, (**G**) A549, (**H**) SW-13, (**I**) Caki-1, (**J**) MIA PaCa-2, (**K**) LS174T, (**L**) SK-MEL-2, (**M**) HeLa, (**N**) PA-1, (**O**) MCF-7, and (**P**) PC-3. Black lines, negative control (control antibody); blue lines, anti-sialoglycoconjugate antibodies (mouse IgM); red lines, anti-sialoglycoconjugate antibodies (mouse IgG). (**Q**) Western blot analysis of the cancer cell lines by 12E3 antibody for polySia–NCAM. P.C., positive control (pig embryonic brain homogenate); MW, molecular size markers.

**Figure 3 ijms-23-05569-f003:**
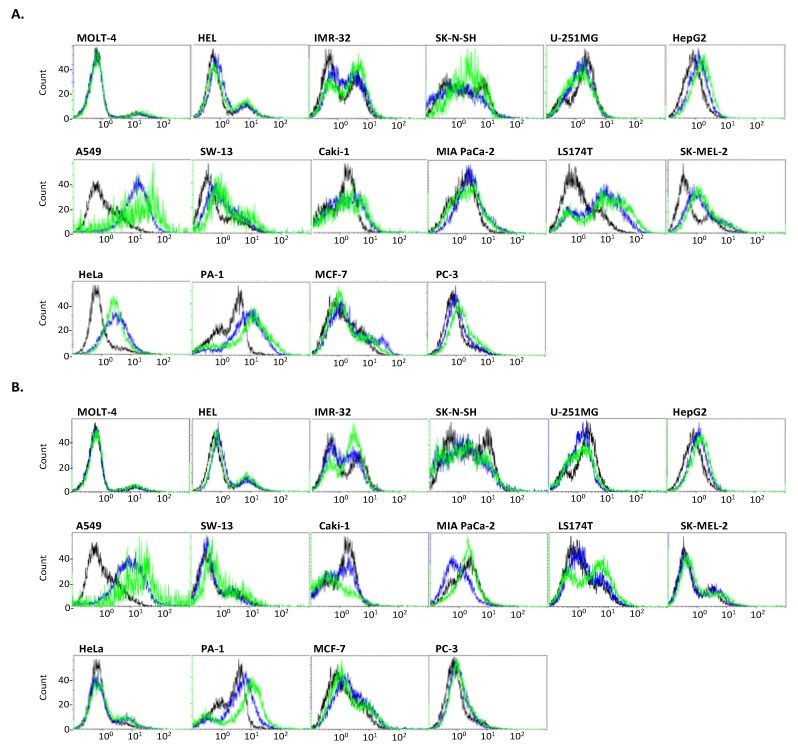
Flow cytometric analysis of cancer cell lines using anti-non-human sialic acid antibodies. In each histogram, black line, negative control (control antibody); blue line, anti-sialoglycoconjugate antibodies (mouse IgM), 2-4B (**A**) and kdn8kdn (**B**). Green line, anti-sialoglycoconjugate antibodies, 2-4B (**A**) or kdn8kdn (**B**), in the presence of 0.1 mM Neu5Gc (**A**) or 0.1 mM Kdn (**B**), respectively.

**Figure 4 ijms-23-05569-f004:**
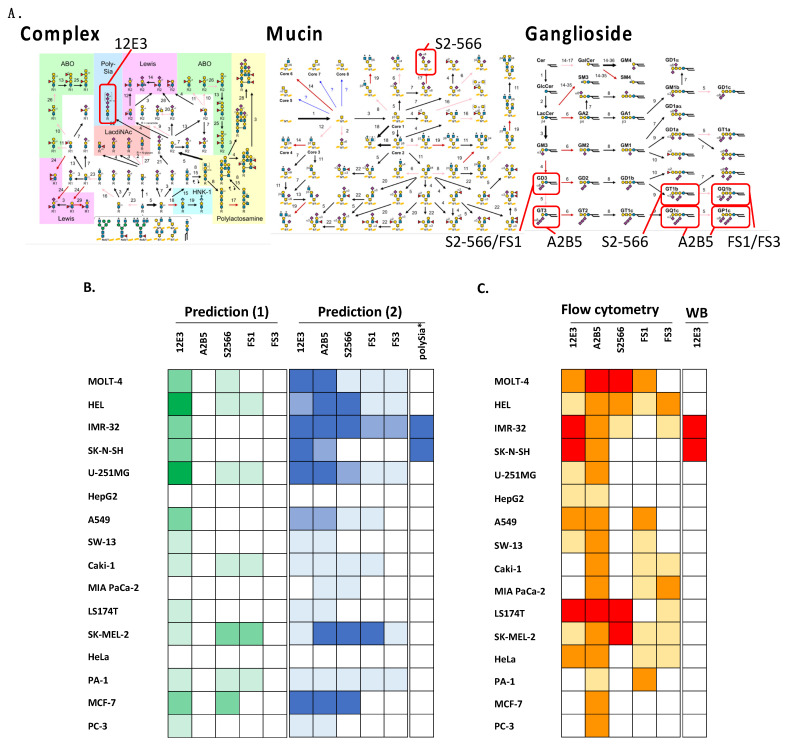
Validation of the results obtained by the dry and wet analyses. (**A**) Epitope structures recognized by the anti-sialoglycoconjugates antibody used in this study. Combinations of the antibody with its epitope structure (red square) are indicated on the pathway maps for the complex N-glycan, mucin, and ganglioside. (**B**) Expression patterns of oligo/polySia structures predicted by (1) the Glycomaple method and (2) our method (Figure 1). (**C**) Expression patterns of oligo/polySia structures observed by the flow cytometry analysis and Western blotting. For B and C, dark color, high expression; light color, low expression; white, no expression.

**Table 1 ijms-23-05569-t001:** Human cancer cell line used in this study.

Cell	Cell Type	Depmap ID ^a^	Note
MOLT-4	T-leukemia	ACH-000964	Variant of MOLT-3
HEL	erythroblast leukemia	ACH-000004	
IMR-32	neuroblastoma	ACH-000310	
SK-N-SH	neuroblastoma	ACH-000149	
U-251MG	astrocytoma	ACH-000232	
HepG2	hepatoma	ACH-000739	
A549	lung adeno carcinoma	ACH-000681	
SW-13	adrenal cortex small-cell carcinoma	ACH-001401	
Caki-1	kidney carcinoma clear cell	ACH-000433	
MIA PaCa-2	pancreatic carcinoma	ACH-000601	
LS174T	colon adenocarcinoma	ACH-000957	Variant of LS180
SK-MEL-2	melanoma	ACH-001190	
HeLa	Cervical cancer	ACH-001086	
PA-1	ovarian teratocarcinoma	ACH-001374	
MCF-7	breast cancer	ACH-000019	
PC-3	prostatic cancer	ACH-000090	

^a^ CELL ID number of depmap.

**Table 2 ijms-23-05569-t002:** Anti-sialoglycoconjugate antibodies.

Antibody	Isotype	Sia Component	DP	Linkage	Epitope	Ref.
12E3	IgM	Neu5Ac	5≤	α2,8	oligo/polyNeu5Ac	[17]
A2B5	IgM	Neu5Ac	3	α2,8	triNeu5Ac, GT3, GQ1c, GP1c, GH1c, OAcGT3	[18,19,20]
S2-566	IgM	Neu5Ac	2	α2,8	diNeu5Ac2-3Gal, GD3, GT1b	[21]
FS1	IgG3	Neu5Ac	2	α2,8	GD3, GQ1b *	[22,23]
FS3	IgG3	Neu5Ac	2	α2,8	GQ1b	[22,23]
2-4B	IgM	Neu5Gc	2≤	α2,8	oligoNeu5Gc	[24]
Kdn8kdn	IgM	Kdn	2–3	α2,8	di/triKdn	[17,25]
3G9	IgM	Sulfated Neu5Ac	-	-	HO_3_S-8*O*-Neu5Ac	[26]

* GQ1b reactivity is much weaker than that of GD3.

## Data Availability

Not applicable.

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
