# Peer review of "Comprehensive Analysis of Oligo/Polysialylglycoconjugates in Cancer Cell Lines"

_ijms, 2022, doi:10.3390/ijms23105569_

Round 1
Reviewer 1 Report
A combined approach of flow cytometry and bioinformatic glycan prediction was performed.
They authors propose anti-sialoglycoconjugate antibodies as powerful tool for the diagnosis of cancers. Otherwise, the pairing between the prediction strategy and the wet results didn't fit well.
The work can be accepted after reviewing the figures. Figure 4a is too small. An English revision is required.
Reviewer 2 Report
In this paper Hane et al. aim at providing a method to predict cell surface glycan structure of cancer cell lines by using gene expression data. The validation of the prediction strategy has been performed by flow cytometry.
Although the aim of the work appears very interesting, the study suffers of a vague clear significance of the results obtained.
Authors should include in the analysis non-cancerous cells as control. This aspect is fundamental to understand whether the glycan patterns observed in cancer cells are discriminative and useful for a diagnostic or therapeutic purpose.
The authors should include a statistical analysis to prove that the predictive approach has a real significance in predicting cell surface glycan strucutures.
In the flow cytometry histograms (figure 2 and 3) the negative control antibody does not behave in the same manner with all cancer cell lines. In some samples it seems to give a staining nearly overlapping the specific antibody staining.
